# Pulmonary fibrosis distal airway epithelia are dynamically and structurally dysfunctional

Ian T. Stancil [1], Jacob E. Michalski[1], Duncan Davis-Hall [2], Hong Wei Chu[3,4], Jin-Ah Park [5], Chelsea M. Magin[2,3,6], Ivana V. Yang[1], Bradford J. Smith [2,7], Evgenia Dobrinskikh[1] & David A. Schwartz [1,8✉]

The airway epithelium serves as the interface between the host and external environment. In many chronic lung diseases, the airway is the site of substantial remodeling after injury. While, idiopathic pulmonary fibrosis (IPF) has traditionally been considered a disease of the alveolus and lung matrix, the dominant environmental (cigarette smoking) and genetic (gain of function *MUC5B* promoter variant) risk factor primarily affect the distal airway epithelium. Moreover, airway-specific pathogenic features of IPF include bronchiolization of the distal airspace with abnormal airway cell-types and honeycomb cystic terminal airway-like structures with concurrent loss of terminal bronchioles in regions of minimal fibrosis. However, the pathogenic role of the airway epithelium in IPF is unknown. Combining biophysical, genetic, and signaling analyses of primary airway epithelial cells, we demonstrate that healthy and IPF airway epithelia are biophysically distinct, identifying pathologic activation of the ERBB-YAP axis as a specific and modifiable driver of prolongation of the unjammed-to-jammed transition in IPF epithelia. Furthermore, we demonstrate that this biophysical state and signaling axis correlates with epithelial-driven activation of the underlying mesenchyme. Our data illustrate the active mechanisms regulating airway epithelial-driven fibrosis and identify targets to modulate disease progression.

[1] Department of Medicine, University of Colorado Anschutz Medical Campus, Aurora, CO, USA. [2] Department of Bioengineering, University of Colorado Denver | Anschutz Medical Campus, Aurora, CO, USA. [3] Department of Medicine, Division of Pulmonary Sciences and Critical Care Medicine, University of Colorado Anschutz Medical Campus, Aurora, CO, USA. [4] Department of Medicine, National Jewish Health, Denver, CO, USA. [5] Harvard T.H. Chan School of Public Health, Boston, MA, USA. [6] Department of Pediatrics, University of Colorado Anschutz Medical Campus, Aurora, CO, USA. [7] Department of Pediatrics, Division of Pediatric Pulmonary and Sleep Medicine, University of Colorado Anschutz Medical Campus, Aurora, CO, USA. [8] Department of Immunology and Microbiology, University of Colorado Anschutz Medical Campus, Aurora, CO, USA. ✉email: David.Schwartz@cuanschutz.edu

From development to disease, the airway epithelium receives, integrates, and responds to local and environmental signals[1–7]. This dynamic nature of the airway epithelium has been appreciated during branching morphogenesis, asthma pathogenesis, and airway regeneration[3–11]. This incorporation of and response to autonomous and nonautonomous signals are critical for tissue homeostasis[12–16]. When these response mechanisms are diminished or aberrantly activated diverse pathologies can arise.

Idiopathic pulmonary fibrosis (IPF) is characterized by progressive scaring of the lung, destruction of the alveolus, and bronchiolization of the distal airspaces[17]. Our understanding of disease-specific fibroblast and alveolar epithelium contributions to IPF pathogenesis have advanced significantly[18–20]. However, the airway-related changes continue to be poorly understood even with significant structural, signaling, and genetic contributions arising from the airway epithelium[21–26]. Recent work has demonstrated the importance of the airway epithelium in furthering pathogenesis after bleomycin in models of fibrosis;[27,28] however, the full scope and functional ramifications of airway dysfunction in IPF remain unclear.

Previous groups have investigated epithelial function through a biophysical lens, interrogating the collective properties of the tissue[29–31]. One framework to understand the epithelial collective is through the jamming transition. The jamming transition is a physical transition from migratory (unjammed) to non-migratory (jammed) in the absence of wounding or epithelial cell-type changes and encodes information about tissue structure[32]. The jamming transition has been demonstrated in vivo as a modality for organismal development and in vitro for cancer metastasis and asthma pathogenesis[33–37]. This tissue-level migratory process provides a conceptual and experimental framework to understand how the airway epithelium can lead to large-scale remodeling and promote fibrosis. To address this unknown we utilized the well-established air-liquid-interface (ALI) culture system to investigate primary airway epithelial cells differences from healthy and IPF patients.

Here we identify that culture of distal airway epithelia recapitulate in vivo airways and have a significantly delayed jamming transition. This extended unjammed phase was associated with a failure to properly moderate ERBB-YAP signaling. This ERBB-YAP axis was identified as both necessary and sufficient for driving the jamming transition in distal airway epithelia. Furthermore, we identified that this signaling modality dynamically interacts with the most significant genetic risk variant (*MUC5B*) to potentiate the unjammed phase and is specific to IPF when compared to other etiologically similar diseases. Finally, we identified that this distal airway epithelial dysfunction was sufficient to drive primary lung fibroblast activation but could be rescued through modulation of distal airway epithelial jamming and aberrant signaling cessation. Taken together, these findings highlight a novel perspective in IPF, whereby the distal airway epithelia act as a driver of remodeling and fibrosis, and not a secondarily impacted tissue.

## Results

**Jamming is dysregulated in distal IPF epithelia**. Healthy and IPF lungs display significant differences in KRT5 + basal and MUC5B+ mucus producing cells in the distal airways[2,5,21–25,38] (Fig. 1a). To understand this further, we obtained primary human bronchiolar epithelial cells from distal airways (<2 mm internal diameter) from bronchial brushes in individuals without a history of lung disease or individuals with a diagnosis of IPF (Supplementary Table 1). These cells were cultured at ALI for 14 days until a well-differentiated pseudostratified epithelium was

achieved (Fig. 1b). We found that epithelial cells from patients with IPF exhibit increased expression of KRT5 and MUC5B, indicating in vitro cultures of distal airway epithelia recapitulate the in vivo airway (Fig. 1c). In addition, during differentiation, control and IPF epithelia present distinct differences in cellular differentiation (MUC5B+ cells), composition (MUC5B+ and KRT5+ cells), and epithelial integrity (transepithelial electrical resistance (TEER)), but not in proliferation (Ki67) or basal cell subtypes (KRT5, KRT8 and P63+ cells) (Supplementary Fig. 1 A–E).

To understand how disease status alters this epithelial layer, we investigated whether the jamming transition was altered in IPF epithelia. The jamming transition is a tissue-wide phase transition from a fluid-like (unjammed/motile) to a solid-like (jammed/non-motile) phase that has been implicated in vertebrate body-axis elongation, cancer metastasis, and asthma pathogenesis[31–37]. This jamming transition from a motile to non-motile phase occurs without epithelial dedifferentiation or changes in cellular density[31–37]. This fluid-to-solid transition occurred by day 8 of ALI for control cells, while IPF cells persisted in the fluid phase past day 14 of ALI. (Fig. 1d and Supplementary Movie 1–4). These migratory differences were quantified via cellular mean-squared displacement (MSD) and an overlap parameter (Q), demonstrating a prolonged migratory phenotype present in IPF distal airway epithelia (Fig. 1e, f). Structural measurements of the epithelia validate the occurrence of a jamming transition in control cells by day 8 with an average perimeter-to-area cell-border ratio ($q = \frac{\text{cell perimeter}}{\sqrt{\text{cell area}}}$) below the predicted transition value ($p_0 = 3.813$)[31,32,36,37] (Fig. 1g). Concordantly, the IPF epithelia maintain an elongated cell shape accompanied with higher migratory speeds over the course of ALI cultures, consistent with a delay in the jamming transition (Fig. 1h). This prolonged unjammed phase in IPF could be indicative of a persistent disruption in epithelial homeostasis.

We next asked whether airway epithelial dysfunction in IPF lungs was anatomically restricted. Epithelial cells from proximal (>10 mm diameter) and distal (<2 mm diameter) airways were dissected from the same patient with IPF and followed through monolayer maturation. Proximal and distal airways diverged around day 9 of ALI with proximal airways undergoing the jamming transition similar to control distal airway epithelia, while IPF distal airways persisted in the unjammed phase past day 14 of ALI (Fig. 1i). These migratory differences were validated by MSD and Q confirming the persistence of the unjammed phase only in IPF distal airway epithelial cells (Fig. 1j, k). In addition, in distal epithelia, the cell shape index was consistent with higher migratory speeds (Fig. 1l, m). These data indicate that the jamming transition is temporally preserved in IPF proximal airways but dysregulated in the distal airways.

**ERBB-YAP activation is sufficient to induce unjamming**. To identify gene and signaling pathways related to delayed jamming, we performed bulk RNA-sequencing of distal airway epithelial cells from control and IPF donors at various ALI days across the jamming transition (Supplementary Fig. 2A). We identified 38 genes associated with the unjammed phase by comparing genes downregulated during the jamming transition, between days 4 and 8 in control cells, to genes upregulated at days 8 and 14 in IPF cells (Supplementary Fig. 2B, C and Supplementary Table 2). KEGG analysis of these 38 differentially expressed genes demonstrated an enrichment for MAPK signaling, ERBB signaling, and transcriptional regulation of cancer. Gene ontology analysis (biological process and molecular function) demonstrated an enrichment for genes involved in ERBB and growth factor signaling (Supplementary Fig. 2D). A network analysis of these conserved genes

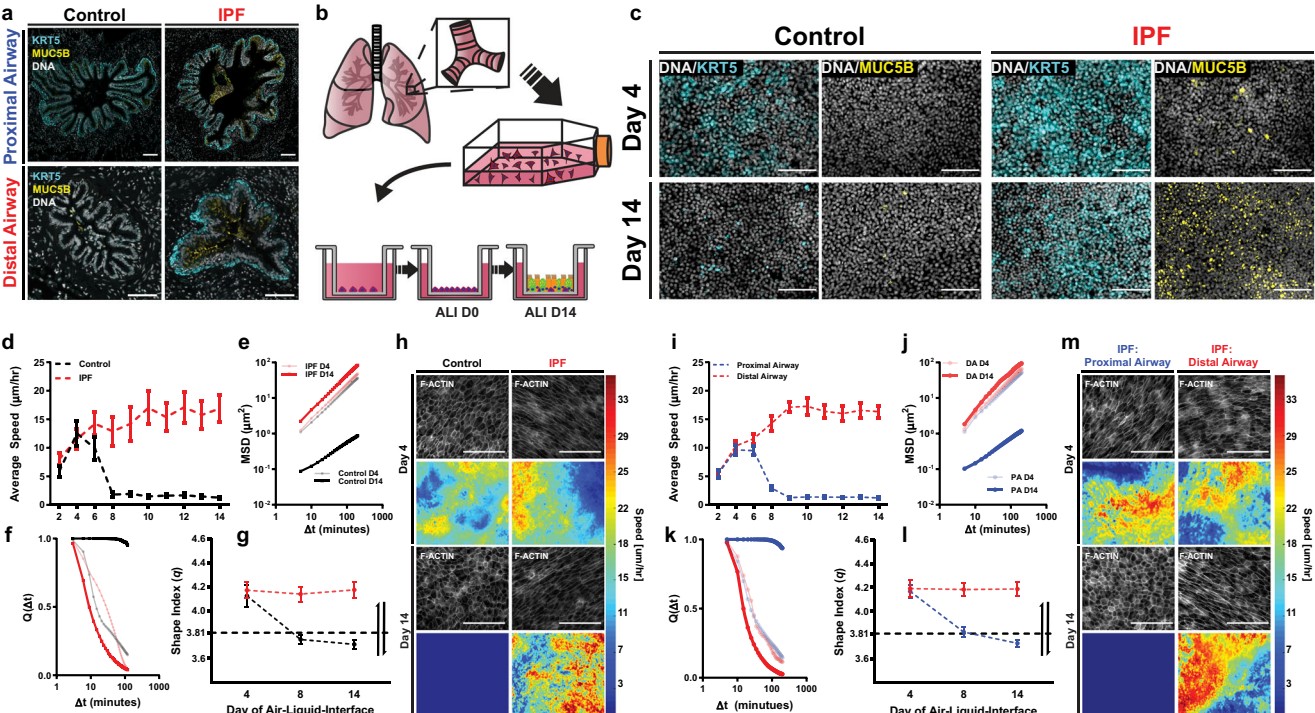

**Fig. 1 The IPF distal airway is recapitulated in vitro and persists in the unjammed phase.** Representative distal human airways have increased KRT5 and MUC5B protein (**a**) when compared to their proximal airway or control counterpart. Primary human bronchiolar epithelial cells were isolated via (**b**) bronchial brushing (distal airways from 6th to 7th generation, generally <2 mm diameter) or CT-guided isolations (proximal >10 mm diameter and distal < 2 mm diameter) and expanded from passage 0 to 2 prior to experimentation. Distal epithelia from control and IPF bronchial brushes were seeded onto porous Transwells and stained for (**c**) KRT5 or MUC5B, 4 or 14 days after establishing ALI demonstrating a similar phenotype to the in vivo patient airway. Control and IPF distal epithelia were imaged for 3 h during epithelial maturation. Utilizing particle image velocimetry (PIV – see methods), cellular speeds (**d**), mean-squared displacement (MSD) (**e**), and the overlap parameter (Q) (**f**), illustrate a biophysical divergence between control and IPF epithelia between days 6 and 8 of ALI. The perimeter-to-area ratio ($q$) of segmented epithelia (see methods) from control distal epithelia fell below the predicted jamming transition threshold ($p_0 = 3.81$) around day 8 of ALI, whereas IPF epithelial maintain an elongated cell body through day 14 (**g**) correlating with the persistence of a migratory, or unjammed state. Representative F-actin staining and speed maps (**h**) of control and IPF epithelia illustrate the biophysical differences in speed and cell shape temporally (scale bar = 100 $\mu$m). Utilizing cells from CT-guided dissections, we seeded IPF epithelia from proximal or distal airways from within the same patient. Time-lapse imaging for 3 h demonstrated that this unjammed state, assessed via speed (**i**), MSD (**j**), and Q (**k**), and q (**l**) was specific to the distal airways of IPF patient samples with proximal airways entering the jammed phase ~day 9 of ALI. This phenotype was appreciated by representative F-actin-stained epithelia and speed maps (**m**) illustrating the biophysical differences of intra-airway epithelia. Shown: mean ± 95% confidence interval for $n = 4$ donors (control epithelia) and $n = 3$ (IPF epithelia) with ≥3 replicates per donor and 100 μm scale bar.

revealed protein-protein interactions between epidermal growth factor receptor (EGFR) and the inducible transcriptional co-activator, YES-associated protein (YAP) (Supplementary Fig. 2E). These findings are distinct from the previously reported RNA-sequencing analysis in bronchial epithelial cells during the jammed and unjammed phase. Previous analyses have demonstrated significant enrichment for cytoskeletal reorganization, stress fiber formation, and extracellular matrix suggesting a dynamic interplay of downstream pathways. However, our analysis demonstrated a robust enrichment for ERBB and HIPPO/YAP, suggesting that delayed jamming as a result of epithelial differentiation differs from the unjammed phase induced by an external stimulus, such as mechanical stress[39,40].

We validated our RNA findings in vivo and in vitro identifying increased amphiregulin mRNA (AREG) in IPF distal airways (Fig. 2a) and increased in ERBB2 protein and YAP nuclear localization in cultured IPF epithelia[41–43] (Fig. 2c, d). In both control and IPF distal epithelia, expression of ERBB family receptors and YAP target genes were higher in the unjammed phase and decreased in control epithelia during the jamming transition (Fig. 2e–h). This inverse correlation between jamming and ERBB-YAP motivated the investigation of whether ERBB or

YAP activation could induce unjamming in control epithelia. Control cells (on ALI day 14) were treated with a high-affinity (EGF or TGFα) or low-affinity (AREG) EGFR ligands[44]. We found that only AREG treatment had the ability to induce an unjammed phase in control epithelia without diminishing barrier function (Supplementary Fig. 3A–F). In addition, a small molecule (XMU-MP-1) inducer of YAP nuclear localization had the ability to also stimulate unjamming (Supplementary Fig. 3G, H). This AREG- and XMU-induced unjamming persisted longer than 48 h (Fig. 2l, m and Supplementary Movie 5–7). Further, neither AREG nor XMU treatment resulted in increased proliferation, but both did lead to YAP nuclear localization (Supplementary Fig. 3I–L). EGFR and YAP activation are frequently associated with an epithelia-to-mesenchymal transition (EMT)[45–47], which is a common modality of collective migration in epithelial layers[48–50]. However, in our system, long-term treatment with EGFR or YAP activators did not elicit an EMT or partial-EMT phenotype when compared to TGFβ1 treatment (Supplementary Fig. 4A–H). This finding is consistent with previous work demonstrating that unjamming in well-differentiated airway epithelia is dynamically, structurally, and functionally distinct from the EMT[32].

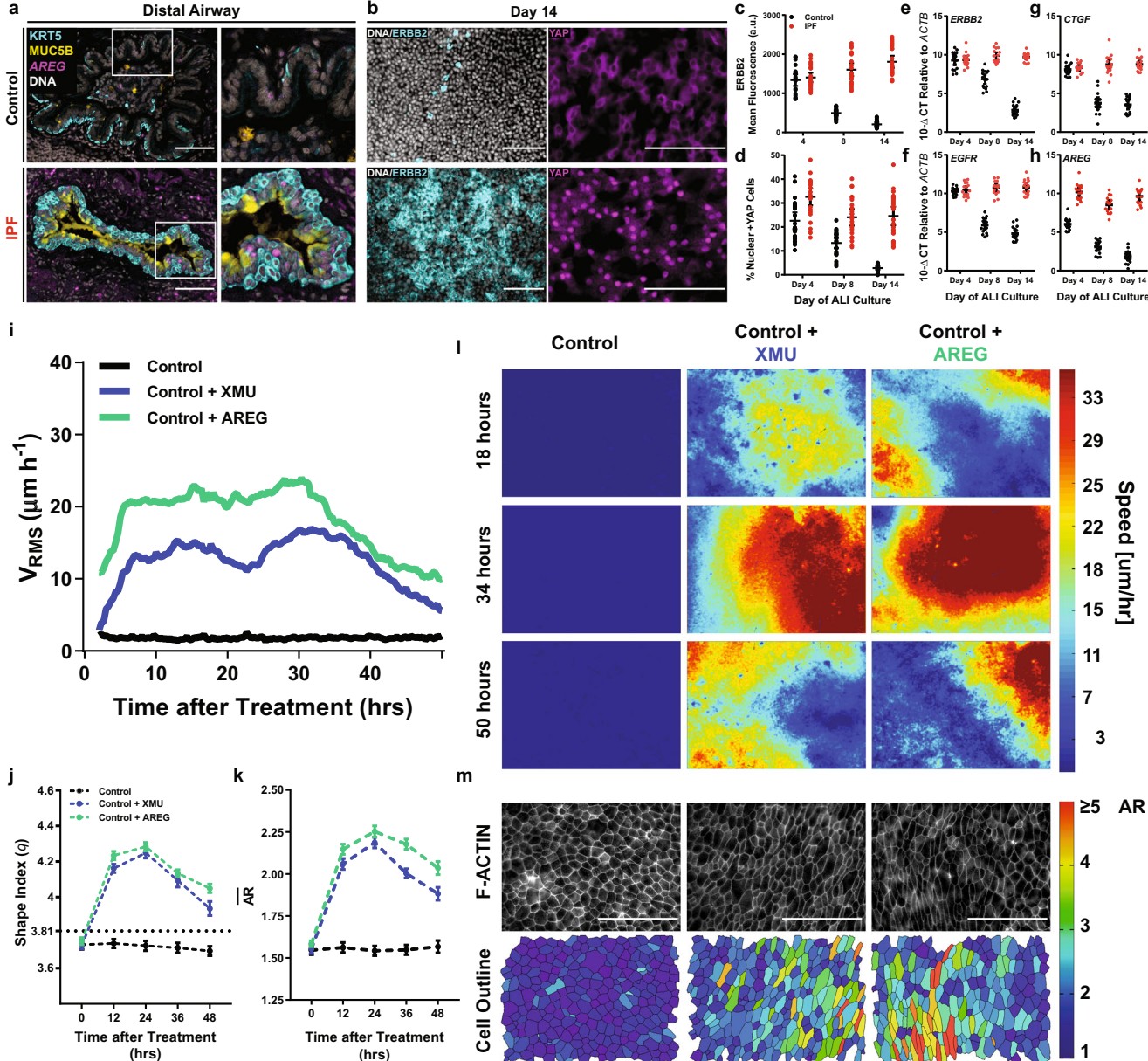

**Fig. 2 EGFR and YAP activation induce unjamming in airway epithelia.** Representative distal airway epithelial cells were stained for KRT5 and MUC5B protein, and *AREG* mRNA in control or IPF (**a**) patient samples with increased *AREG* present in the IPF epithelium. Representative images of control or IPF distal epithelia stained for ERBB2, or YAP (**b**) demonstrate increases in ERBB2 protein and YAP nuclear localization in IPF distal epithelia. This difference in ERBB protein (**c**) and YAP nuclear localization (**d**) was most pronounced at later ALI days with protein/localization differences becoming most apparent as control distal epithelia entered the jammed state. These protein level changes were also accompanied by significant gene level changes in ERBB family receptors *EGFR* (**e**) and *ERBB2* (**f**) as well as YAP target genes *AREG* (**g**) and *CTGF* (**h**) with control epithelial differences becoming pronounced on day 8, across the jamming transition. Control epithelia from distal airways on day 14 of ALI were treated with XMU (a small molecule YAP activator) or AREG (an EGFR ligand) to elicit unjamming of the static, quiescent epithelium. Upon treatment, XMU and AREG elicited a collective migratory phenotype as measured by root-mean-squared velocity $V_{RMS}$ (**i**). This induced unjammed state was also quantified via cell shape changes with the perimeter-to-area (q) (**j**) and aspect ratios (**k**), cells were analyzed every 12 h after treatment was initiated. Representative speed maps taken every 16 h after treatment (**l**) and cell boundary segmentation at 48 h after treatment (**m**) demonstrated concordant migration and cell body elongation consistent with the induction of an unjammed state. Shown: mean ± 95% confidence interval for $n = 4$ donors (control epithelia) and $n = 3$ donors (IPF epithelia) with ≥2 replicates per donors and scale bars representing 100 μm.

After XMU or AREG treatment on ALI day 14, the migratory capacity of control epithelia begins to diminish at ≈36 h post-treatment. However, we found that combined treatment of AREG and XMU could sustain this migratory phenotype longer than 48 h (Supplementary Fig. 5A–E). We also treated control epithelia on ALI day 28 and found that this unjammed phase was induced regardless of monolayer maturity (Supplementary Fig. 5F–J). Interestingly, after XMU or AREG treatment, YAP nuclear localization was predominantly in multi-ciliated cells (FOXJ1+) (Supplementary Fig. 6A–G). Staining of IPF lung tissue revealed FOXJ1+/YAP+ co-positive populations in distal airways in regions of minimal fibrosis (Supplementary Fig. 6H).

**ERBB-YAP axis specifically regulates IPF unjamming.** Next, we sought to understand if the pathological unjammed phase in IPF

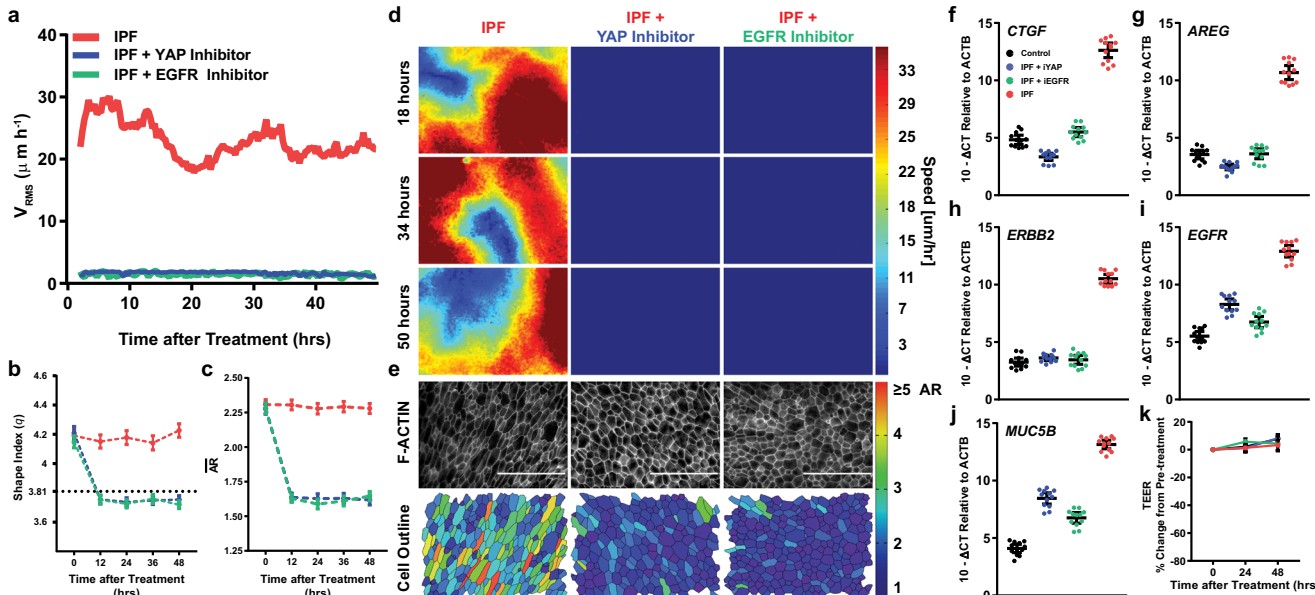

**Fig. 3 Inhibition of EGFR and YAP induce jamming in IPF distal epithelia.** Distal airway IPF cells were treated at day 14 for ALI with Verteporfin (YAP inhibitor) or AG1478 (EGFR inhibitor) (**a**) and filmed continuously for 48 h demonstrating a cessation of collective migration as assessed by $V_{RMS}$. Cell shape ratio measurements, perimeter-to-area (**b**) and aspect (**c**) were taken every 12 h after treatment initiation and illustrated the acquisition of a cobblestone-like phenotype which persisted for the entire duration of the measurement (up to 48 h). Representative speed maps taken every 16 h after treatment (**d**) and cell boundary segmentation at 48 h after treatment (**e**) demonstrate a concordant cessation in migration and acquisition of a cobblestone-like apical surface, color coded by aspect ratio (AR) of the epithelium consistent with a jammed state. IPF epithelia treated with Verteporfin or AG1478 had reduced YAP target gene expression AREG (**f**) and CTGF (**g**) as well as ERBB family receptor ERBB2 (**h**) and EGFR (**i**) 48 h after treatment, nearly back to control epithelial levels. In addition, gene expression of MUC5B (**j**) was also decreased after treatment. Inhibition of YAP or EGFR did not significantly impact barrier function as assessed via TEER (**k**) with marginal increased barrier function occurring after treatment. Shown: mean ± 95% confidence interval for $n = 3$ donors (control epithelia) and $n = 4$ donors (IPF epithelia) with ≥2 replicates and scale bars representing 100 μm.

distal epithelia could be rescued through inhibition of these targets (EGFR or YAP). Utilizing IPF cells on day 14, we treated cultures with AG1478 (EGFR inhibitor) or Verteporfin (YAP inhibitor). Treatment with either inhibitor halted migration and induced a cobblestone-like cell shape, indicating a shift in the biophysical state of the cells from unjammed-to-jammed (Fig. 3a–e and Supplementary Movie 8–10). In addition, this targeted inhibition reduced the transcription of ERBB family receptors, YAP target genes, and MUC5B nearly back to control levels (Fig. 3f–k)

EGFR activation can signal through myriad of downstream effectors. We sought to understand the specificity of the EGFR activation in unjammed IPF epithelia and whether other downstream targets had the ability to induce jamming (Supplementary Fig. 7A). Treatment with MK-2206 (AKT 1/2/3 inhibitor), Stattic (STAT3 inhibitor), Ruxolitinib (JAK1/2 inhibitor), or Sotrastaurin (pan-PKC inhibitor) did not induce the jamming transition in distal IPF epithelia (Supplementary Fig. 7B, C). Taken together, these findings indicated that the prolonged unjammed phase induced by EGFR and YAP in IPF epithelia is likely not due to other downstream EGFR targets (i.e., AKT1/2/3, STAT3, JAK1/2, PCK), but instead is specific to the ERBB-YAP axis. ERBB-YAP signaling axis occurs via a PI3K-PDK1 directed mechanism which disrupts the ability of LATS1/2 to retain YAP in the cytoplasm (Supplementary Fig. 8A)[51–54]. To determine the signaling requirements of EGFR-YAP prolonged unjamming in IPF, we utilized IPF distal epithelial cells on ALI day 14 and tested the ability of specific inhibitors in the EGFR-YAP axis to induce jamming. Inhibition of the intermediate signaling effectors in the EGFR-YAP axis induced jamming, which when followed persisted for >144 h after only the initial treatment (Supplementary Fig. 8B–U). This inhibition was accompanied by

concomitant exclusion of YAP from the nucleus and concordant gene expression (Supplementary Figs. 9 and 10). These data demonstrate the requirement of the EGFR-YAP signaling axis in maintaining a prolonged unjammed phase in IPF epithelia.

**Unjamming is responsive to genetic risk variants and is disease specific.** We next wanted to understand whether known genetic risk factors specific to IPF affected AREG- or YAP-driven unjamming. The gain-of-function MUC5B promoter variant rs35705950 is the dominant genetic IPF risk variant, which results in MUC5B overexpression in the distal airway epithelium[21,55]. We treated control distal epithelia from individuals with no copies (GG) or one copy (GT) of the MUC5B promoter variant with AREG or XMU. Untreated distal epithelia persisted in the jammed phase regardless of their MUC5B status, however, GT epithelia treated with AREG had a higher root-mean-squared velocity than their GG counterparts (Supplementary Fig. 11A, B). In addition, GT epithelia treated with AREG persisted in the unjammed phase significantly longer than GG epithelia (Supplementary Fig. 11C–E). This genotype-dependent unjamming did not alter barrier function and was confirmed via cell shape measurements. These data demonstrate a MUC5B-driven differential response to AREG-induced unjamming in airway epithelia. A possible explanation for this finding is a shift in cellular metabolism due to increased MUC5B expression from the promoter variant as well as ERBB activation. Alternations in cellular metabolism have been demonstrated previously in the unjammed phase, cytoskeletal reorganization, and endoplasmic reticulum stress[56–60].

We then asked whether EGFR-YAP dependent unjamming was specific to IPF epithelia or if this axis was common among other chronic lung disease, such as chronic obstructive

pulmonary disease (COPD). While COPD distal epithelial cells persisted in an unjammed phase past day 14, inhibition of the EGFR-YAP axis failed to induce jamming (Supplementary Fig. 12A–J). These data suggest that EGFR-YAP driven unjamming is specific to IPF, and that COPD has a non-EGFR-YAP dependent mechanism regulating the unjammed phase.

**Dynamics and signaling regulate epithelial-driven fibrosis.** Next, we evaluated whether IPF distal epithelia could induce a pro-fibrotic response in human lung fibroblasts (HLF). We found that AREG secretion into the basal media was increased in IPF, and unjammed control distal epithelia (Supplementary Fig. 13A, B) and that this secretion was decreased in IPF distal epithelia after pharmacological jamming (Supplementary Fig. 13C). This finding raised the question whether AREG could drive a pro-fibrotic state in fibroblast. To pursue this, we seeded healthy HLFs onto physiologically soft (3 kPa) hydrogels to prevent artificial activation of the HLFs (Supplementary Figs. 14A–D and 15)[61–64]. HLFs were treated with conditioned media from control (jammed), control + YAP (unjammed), control + AREG (unjammed), and IPF (unjammed) distal epithelial cultures (Fig. 4a, b). Treatment of HLF with conditioned media from control-unjammed or IPF-unjammed distal epithelia resulted in increases in the total number of fibroblasts and markers of activation when compared to control untreated epithelia (Fig. 4c–e). This treatment also induced significant gene expression changes in canonically pro-fibrotic genes (Supplementary Fig. 16A–O). These data suggests that the pathologic unjamming of the epithelia activates fibroblasts in the surrounding mesenchyme consistent with other mechanical manipulations of airway epithelia[65].

We then asked whether AREG was the primary driver of this fibrotic response. We treated HLFs with IPF conditioned distal epithelial media supplemented with TGFBRI inhibitor (SB431542), AREG neutralizing antibody (iAREG), or SB431542 + iAREG (Fig. 4f) and found that treatment reduced fibroblast activation (Fig. 4g–j). However, none of these treatments alone were able to fully rescue the increase in cell number or the associated pro-fibrotic gene expression (Supplementary Fig. 17A–O). Interestingly, AREG inhibition induced a more significant differentiation-related rescue than TGFBRI inhibition alone (Fig. 4h–j). This supports our hypothesis that the epithelial-driven fibrotic response seen in our system is multifactorial and that AREG plays a substantial role.

Next, we tested if direct inhibition of the epithelia had the ability to more robustly prevent the fibrosis-like phenotype observed. Conditioned media were collected from EGFR-, YAP-, and AREG-inhibited IPF distal epithelia at 48 h after treatment and was used to culture HLFs (Fig. 4k). Treatment significantly reduced HLF differentiation, decreased the total number of cells, (Fig. 4l–o) and reduced HLF gene expression of canonically pro-fibrotic targets (Supplementary Fig. 18A–O). Notably, direct targeting of unhealthy distal epithelia provides a more substantial rescue when compared to targeting fibroblasts.

## Discussion
Our results indicate that IPF distal airway epithelium plays an active role in driving fibrosis and is dynamically and structurally distinct from normal airway epithelia. Further, the ERBB-YAP axis provides a modifiable cascade to reverse the persistent biophysical defects of IPF epithelia. The unjammed phase represents a biophysical property by which collective migration of epithelial layers can traverse large distances. Future studies are needed to determine if this migratory state is implicated in the structural airway epithelial remodeling observed in IPF, including honeycomb cyst formation, bronchiolization of the distal airspace, and

loss of terminal bronchioles. Taken together, we propose that the biophysical properties of the airway epithelium are dynamic and actively participate in disease progression integrating signaling and genetic causes of pulmonary fibrosis.

## Methods
**Antibodies.** For immunofluorescence, we used the following antibodies: rabbit monoclonal anti-ERBB2 (Cell Signaling, 2165, 1:250), chicken polyclonal anti-KRT5 (BioLegend, 905901, 1:500), mouse monoclonal anti-Ki67 (Cell Signaling, 9449, 1:500), mouse monoclonal anti-MUC5B (Novus Biologics, NBP2-50390, 1:5000), rabbit monoclonal anti-Vimentin (Cell Signaling, 5741, 1:100), rabbit monoclonal anti-YAP (Cell Signaling, 14074, 1:250), mouse monoclonal anti-FOXJ1 (Invitrogen, 14-9965-82, 1:500), goat polyclonal anti-FOXJ1 (R&D Systems, AF3619, 1:500), mouse monoclonal anti-MUC5AC (Invitrogen, MA5-12178, 1:200), rat monoclonal anti-SCGB1A1 (R&D, MAB4218, 1:500), rabbit monoclonal anti-KRT8 (Abcam, ab53280, 1:100), rabbit monoclonal P-63α (Cell Signaling 13109, 1:500), mouse monoclonal anti-alpha smooth muscle actin (Abcam, ab7817, 1:500), Phalloidin-iFluor (Abcam, ab176753, 1:2000). All secondary antibodies were purchased from ThermoFisher and used at a concentration of 1:500. For western blotting, we used the following antibodies: rabbit monoclonal anti-E-cadherin (Cell Signaling, 3195, 1:1000), rabbit monoclonal anti-N-cadherin (Cell Signaling, 13116, 1:1000), rabbit monoclonal anti-Snail (Cell Signaling, 3879, 1:1000, and goat polyclonal anti-beta-actin (Abcam, ab8229, 1:500). All secondary antibodies were purchased from LI-COR and used at a concentration of 1:10,000.

**Immunofluorescence staining.** Cells were fixed with 4% paraformaldehyde in PBS with calcium and magnesium for 30 min at room temperature. Cells were then washed with PBS and subsequently permeabilized (Triton-X, at 0.1%) and blocked with 5% bovine serum albumin in PBS for 1 h at room temperature. Cells were then incubated for >16 h at 4 degrees with primary antibodies. Primary antibodies were aspirated, and cells were washed with PBS + 0.1% Tween-20 at room temperature. Cells were then incubated for 1 h at room temperature with secondary antibodies (1:500) and/or Phalloidin-iFluor 488 1:2000. Cells were stained with 4′,6-diamidino-2-phenylindole at 1:20000 (BioLegend, 422801). Subsequently, transwell membranes, or hydrogels, were removed from the well and mounted on glass slides with Fluoromount-G (SouthernBiotech, 0100-01). Slides were then visualized using an Olympus BX63 microscope (Olympus, Tokyo, Japan).

**Epithelial cell culture.** Primary human bronchiolar epithelial cells were obtained from two sources[1] provided by Dr. Hong Wei Chu at National Jewish Health (IRB protocol HS-2604, approved by National Jewish Health Institutional Review Board), or[2] lung transplant collection at the University of Colorado Hospital (IRB protocol: 11-1664 and 18-0572, approved by Colorado Multiple Institutional Review Board). All patients expressed informed consent before cell collection. Epithelial cells from bronchial brushes were obtained from 6th to 7th airway generations, where the airway diameter is approximately ≤2 mm. CT-guided dissection isolated airway cells from proximal > 10 mm or distal < 2 mm airways. Cells from either isolation method were expanded to passage 2 in the same method/media prior to being cultured at ALI. Control (N = 11), IPF (N = 6), and COPD (N = 3) were age matched and controls had no history of chronic lung disease (Supplementary Table 1). Control (N = 11), IPF (N = 7), and COPD (N = 3) were age matched and controls had no history of chronic lung disease (Supplementary Table 1). Cells were seeded on 24-Transwell plates (Corning 3470) coated with type I collagen (Corning, 354236) and maintained under submerged conditions for 5–7 days until the cells reached confluence. Culture media was a 1:1 mixture of DMEM and BEBM (Lonza, CC170) supplemented with retinoic acid (Sigma, R-2625) nystatin (Sigma, N1638), and bovine serum albumin (FisherSci, BP9703100). Once confluent, apical media was removed and maintained at ALI. Distal epithelia began recapitulating the in vivo airway, with production of mucus (6–9 days after establishing ALI) and development of a pseudostratified epithelium. Cells were switched to a starvation media (lacking supplemental epidermal growth factor, hydrocortisone, and bovine pituitary extract) 24 h prior to experimentation.

**Inhibitors used and epithelial inhibitor treatment.** The following compounds were utilized to assess modulation of cellular speed: Tyrphostin AG-1478 (Millipore Sigma, T4182-5MG, 100 nM) and Erlotinib (Selleckchem, S7786, 100 nM) selective EGFR inhibitors; Mubritinib (Selleckchem, S2216, 100 nM) a selective ERBB2 inhibitor; Lapatinib (Selleckchem, S2111, 100 nM) an EGFR/ERBB2 dual inhibitor; OSU-03012 (Selleckchem, S1106, 1 μM) and GSK2334470 (Selleckchem, S7087, 1 μM) selective PDK1 inhibitor; LY294002 (Selleckchem, S1105, 5 μM) a selective PI3K inhibitor; MK-2206 (Selleckchem, S1078, 1 μM) a selective ATK1/2/3 inhibitor; Stattic (Selleckchem, S7024, 5 μM) a selective STAT3 inhibitor; Ruxolitinib (Selleckchem, S1378, 5 μM) a selective JAK1/2 inhibitor; Sotrastaurin (Selleckchem, S2791, 1 μM) a selective PKC inhibitor; Verteporfin (Tocris, 5305, 1 μM) a selective YAP inhibitor; XMU-MP-1 (Tocris, 6482, 1 μM) a selective MST1/2 inhibitor; SB431542 (Tocris, 1614, 5 μM) a selective TGFBRI inhibitor; Amphiregulin neutralizing antibody (R&D, MAB262, 30 μg/ml) Recombinant AREG (R&D, 262-AR), EGF (R&D, 263-EG), and TGF-α (R&D, 239-A-100) EGFR

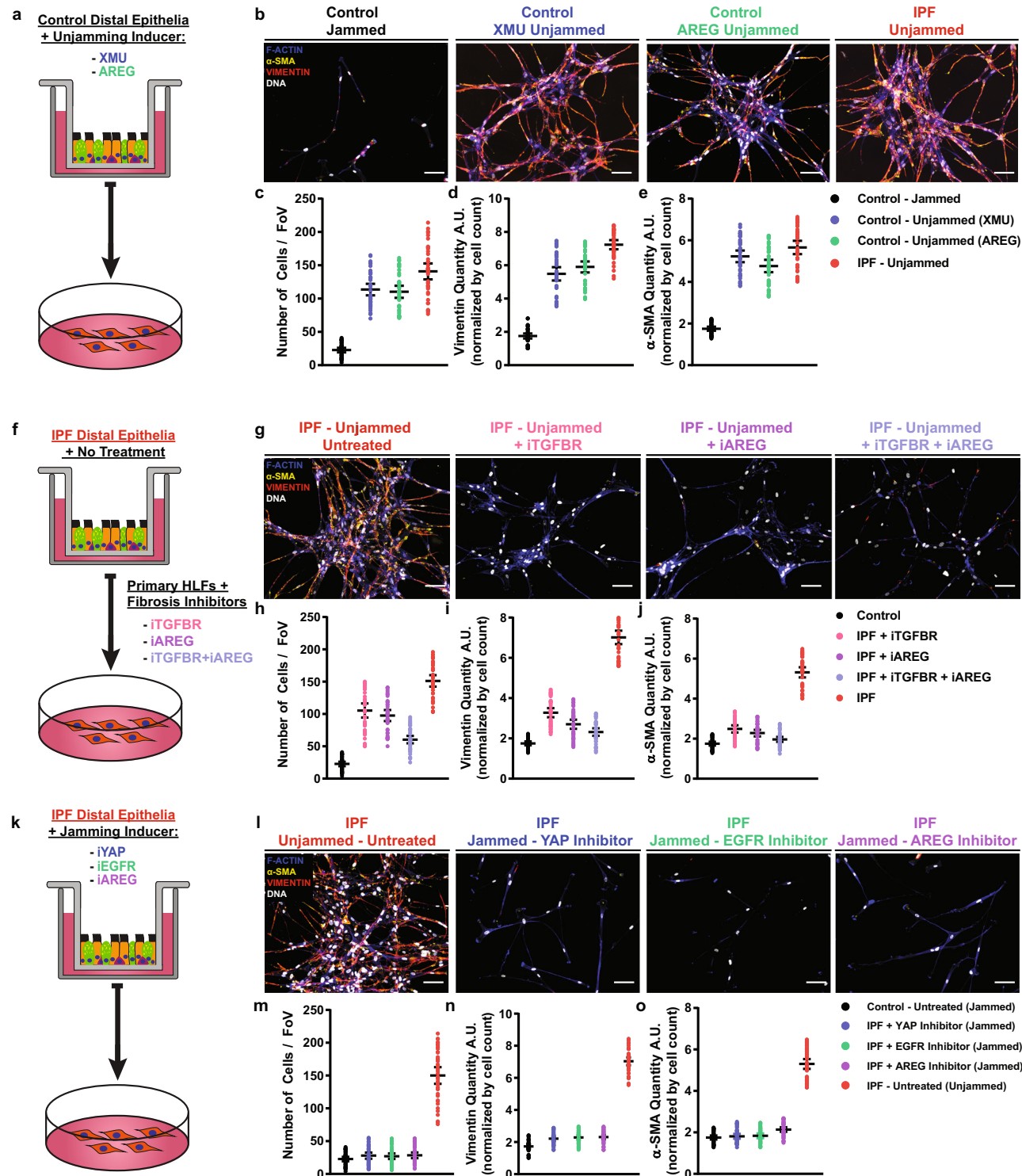

agonists. For inhibitor, activator, and ligand treatments distal epithelia were cultured in a starvation media in the absence of EGF, hydrocortisone, and bovine pituitary extract for 24 h prior to treatment initiation. Cells were then treated for 2–4 h (with exception of AREG neutralizing antibody which was maintained in the culture media), media was aspirated, the wells were washed with warm PBS and then replaced with fresh starvation media without inhibitor/activator/ligand present, and subsequently imaged. For experiments that required subsequent media changes (96 h) cells fresh starvation media was changed 48 h afterwards.

**Culture and treatment of HLFs.** Primary HLFs were obtained from control lungs not suitable for transplantation at National Jewish Health (IRB protocol HS-3209, approved by National Jewish Institutional Review Board under non-human subject

research) in contract with the International Institute for the Advancement of Medicine and Donor Alliance. Prior to donation, informed consent to research use was provided by families and/or donors directly. Briefly, pieces of tissue were places into a tissue-culture dish and weighed down by a coverslip. Cells were cultured in DMEM (Corning, 10-013-cv) supplemented with L-glutamine (25-005-cl), penicillin/streptomycin (Corning, 30-002-cl), and 10% fetal bovine serum (ATCC, 30-2020). These fibroblasts were then sub-cultured in t-75 flasks. All HLFs used in this study ($N = 4$) were seeded at passage 3. Experimentally, fibroblasts were seeded onto 3 kilopascal (kPa) hydrogels at a density of 7500 cells/cm$^2$ and cultured for 24 h in DMEM supplemented with L-glutamine, P/S, and 10% FBS. Fibroblasts were then synchronized in DMEM supplemented with L-glutamine, P/S, and 0.1% FBS for 24 h. Fibroblasts were then treated with epithelial starvation media from jammed non-IPF, unjammed non-IPF, or unjammed IPF cells for 72 h.

**Fig. 4 The signaling and biophysical state of epithelia direct fibroblast activation.** Primary human lung fibroblasts (HLF) were seeded on to a 3 kPa hydrogel and treated for 72 h with conditioned media from control distal epithelia (see methods) treated with either XMU or AREG (**a**) to induce unjamming. Representative immunofluorescence images of HLFs stained for F-actin (blue), α-SMA (yellow), or vimentin (red) showed minimal staining in the HLFs treated with control, untreated conditioned media (**b**). However, fibroblast activation became apparent in control epithelial conditioned media treatment of HLFs from XMU or AREG epithelia. These differences nearly reached the level of HLF activation from untreated IPF distal epithelial conditioned media in terms of cell number (**c**), vimentin (**d**), and α-SMA (**e**) quantity. Schematic of direct fibroblast inhibition (**f**) and representative immunofluorescence images of HLFs (**g**) demonstrating that treating HLFs with SB431542, AREG neutralizing antibody, or a combination of the two during treatment with IPF distal epithelial conditioned media showed an attenuation of the ability for IPF epithelial media to induce an intense pro-fibrotic phenotype indicated by cell number (**h**), vimentin (**i**), and α-SMA (**j**) approached the control-like phenotype. Utilizing media from IPF distal epithelia treated with Verteporfin, AG1478, or AREG neutralizing antibody (**k**) demonstrated the ability to prevent fibroblast activation (**l**). Cell number (**m**), vimentin (**n**), and α-SMA (**o**) were all restored to baseline, control-like levels when using conditioned media from inhibited IPF distal epithelia. Shown: mean ± 95% confidence interval for $n = 4$ donors with three replicates and scale bars representing 100 μm.

Conditioned basal media was changed every 24 h. In the case of direct fibroblast inhibition (via SB431541, AREG neutralizing antibody, or a combination of the two inhibitors), HLFs were treated with starvation media from IPF distal epithelia supplemented with the appropriate inhibitor combination and was changed every 24 h in accordance with the experimental design.

**Quantitative RT-PCR analysis.** Total RNA was extracted using the RNeasy Mini kit and quantified by NanoDrop. Reverse transcription was completed using the High-Capacity cDNA Reverse Transcription Kit (Applied Biosystems). Gene expression was analyzed by using TaqMan Advanced Master Mix and TaqMan assays. They were subsequently run in triplicate on the ViiA 7 Real-Time machine. Primer IDs are located in Supplementary Table 3.

**Immunoblotting.** Cells were lysed in 2× Laemmli Sample Buffer (Bio-Rad, 1610737) supplemented with 5% β-mercaptoethanol, and 1× Halt Phosphatase Inhibitor Cocktail (ThermoFisher, 78426) and boiled for 10 min. Protein quantification was completed using BCA assay and 30 μg of protein was added per lane to a 10% or 4–20% sodium dodecyl sulfate polyacrylamide gel (Bio-Rad) and transferred to a polyvinylidene difluoride membrane (Millipore Sigma, IPVH00010). Membranes were then blocked for 1 h at room temperature in Intercept Blocking Buffer (LI-COR, 927-60001). Membranes were incubated overnight at 4 °C with primary antibody and subsequently incubated with secondary antibodies for 1 h at room temperature. Band visualization was completed using the LI-COR Odyssesy CLx system.

**Barrier function measurements.** The TEER was assessed utilizing the Millicell ERS-2 Voltohmeter (Millipore, MERS00002). Briefly, cells were incubated with PBS with calcium and magnesium (at 37° for 20 min prior to barrier measurement). The probe was inserted, and the resultant ohm reading was recorded. The final value was calculated by:
This value was subtracted by 100 (the resistance of a blank well) and multiplied by 0.33 (the surface area of a single well in a 24 well plate). A minimum of three wells were recorded per donor per treatment condition. These values were subsequently averaged per treatment and graphed.

**In situ hybridization.** RNAScope detection mRNA was used to perform in situ hybridization according to the manufacturer's protocol (Advanced Cell Diagnostics, Hayward, CA). Briefly, formalin-fixed paraffin embedded human lungs were cut into 5 μm thick tissue sections. Slides were deparaffinized in xylene, followed by rehydration in a series of ethanol washes. Following citrate buffer (Advanced Cell Diagnostics) antigen retrieval, slides were treated with protease plus (Advanced Cell Diagnostics) at 40 °C for 30 min in a HybEZ hybridization oven (Advanced Cell Diagnostics). Probes directed against *AREG* mRNA and control probes were applied at 40 °C in the following order: target probes, pre-amplifier, amplifier; and label probe for 10 min. After each hybridization step, slides were washed two times at room temperature. mRNA detection was followed by immunofluorescent staining for basal cells (KRT5 positive) and MUC5B protein. Slides were blocked in 5% BSA buffer for 1 h at room temperature, incubated with primary antibodies overnight at +4 °C, stained with appropriate fluorescently labelled secondaries antibodies and counterstained with 4′,6-diamidino-2-phenyl-lindole (DAPI) at 1:20,000 (BioLegend, 422801). Staining was visualized using an Aperio Vectra 8 whole slide scanner using a 20× lens (Leica, Buffalo Grove, IL). For KRT5 and MUC5B detection in human lungs tissues were deparaffinized in xylene, followed by rehydration in a series of ethanol washes. Following citrate buffer antigen retrieval, slides were blocked in 5% BSA buffer for 1 h at room temperature, incubated with primary antibodies overnight at +4 °C, stained with appropriate fluorescently labelled secondaries antibodies and counterstained with 4′,6-diamidino-2-phenylindole (DAPI) at 1:20,000 (BioLegend, 422801). Staining was visualized using Zeiss 780 laser-scanning confocal/multiphoton-excitation fluorescence microscope with a 34-channel GaAsP QUASAR detection unit and non-descanned detectors for two-photon fluorescence (Zeiss, Thornwood, NY).

**AREG protein detection.** Secreted AREG by the distal epithelia was determine by a Human Amphiregulin Quantikine ELISA kit from R&D (DAR00) following manufacturer instructions. Media from control ($n = 4$) and IPF ($n = 4$) distal epithelia were collected at various days of differentiation as well as after treatment with inhibitors/activators. Three independent wells were per donor were used to determine concentration.

**Immunofluorescence and cell number quantification for distal epithelial experiments.** For total fluorescence quantification (i.e., ERBB2), all images were processed in the same manner with the same microscopy settings. To determine mean fluorescence the channel of interest was separated in ImageJ, and the mean gray value was measured for the field of view and normalized by the total number of cells. Cell counting was achieved by taking 40× images and manually counting all nuclei in a field of view (~200–300). The number of cells also expressing the marker of interested were then counted and normalized by the total number of cells per field. For each experiment a biological $n \geq 3$ was used with at least five fields of view per well and at least two wells being counted in total per donor.

**Immunofluorescence and cell number quantification for HLF experiments.** For fluorescence quantification, (i.e., vimentin or alpha-SMA) images were all processed using the same settings. Fluorescence intensity was determined by channel separation and measurement of mean gray value for the field and normalized to total cell number. Cell number was determined by automated counting in ImageJ. For each experiment a biological $n = 4$ was used with at least ten fields of view per hydrogel and three hydrogels counted per donor.

**Live cell imaging and dynamics measurement.** To assess cellular dynamics, time-lapse microscopy and subsequent analysis were completed. Two imaging schemes were utilized:[1] images were taken every 5 min for 200 min during distal epithelial differentiation;[2] images were taken every 20 min for 48, 96, or 144 h after distal epithelial experimental intervention. Phase contrast images were acquired on a Keyence BZ-X810 with a stage incubator (37 °C, 5% CO₂). Dynamic analysis followed established published workflows[32,34–36]. Using phase contrast images instantaneous velocities were mapped by particle image velocimetry utilizing open source PIVlab in MATLAB[66]. Velocity fields were obtained with two passes ($64 \times 64$ and $32 \times 32$ pixel size interrogation window with 50% overlap) with a pixel size of 0.75488 μm using a Fast Fourier Transformation method. Trajectories from the PIV were seeded onto a grid and obtained via forward-integration of subsequent pixels. These values were then used to determine mean-squared-displacement (MSD) and self-overlap order parameter (Q): $\text{MSD}(\triangle t) = \left\langle \left| r_i(t + \triangle t) - r_i(t) \right|^2 \right\rangle$ and

$Q(\triangle t) = N^{-1} \sum_{i=1}^{N} w^i$ [34,35]. Instantaneous room mean-squared velocity ($V_{RMS}$) of the monolayer was obtained as previously published:[34,35] $V_{RMS}(t) = \sqrt{\frac{1}{M} \sum_{j=1}^{M} \left\langle \left| v(t) \right|^2 \right\rangle j}$

**Cell shape analysis.** Cell shape measurements were achieved via cell-border delineation by F-actin immunofluorescence. Images were segmented utilizing SeedWater Segmenter[67]. Utilizing established workflows[32], we segmented images which were subsequently analyzed in ImageJ for the shape index ($q = \frac{\text{Perimeter}}{\sqrt{\text{Area}}}$) and aspect ratio (major divided by minor axis) for all cells within a given field of view. For each experimental condition an $n \geq 3$ was used with at least three independent wells per donor per treatment condition. Images were acquired at 40× with ~300 cells/field of view in each image. At least 3000 cells were segmented for each experimental condition. To color code aspect ratios, we adapted ImageJ ROI Color Coder.

**Human lung specimens.** Lung specimens from patients with IPF and unaffected controls were obtained from the NHLBI Lung Tissue Research Consortium (LTRC; https://ltrcpublic.com/). All protocols were approved by local Institutional Review

Boards and individuals gave written, informed consent to participate (IRB protocol 15-1147, approved by Colorado Multiple Institutional Review Board).

**RNA-Seq preparation and analysis.** Total RNA was extracted using the RNeasy Mini Kit (Qiagen). RNA quality was assessed using the Agilent 2200 TapeStation system. mRNA libraries were prepared from 500 ng total RNA with TruSeq stranded mRNA library preparation kits (Illumina) and sequenced at the averaged depth of 92.4 M reads on the Illumina NovaSeq 6000 (Illumina). RNA paired-end reads were aligned at the transcript level to Ensembl GrCh38 using Kallisto[68] with the average mapping rate 90.1%. 120,425 transcripts were detected in the mRNA dataset using Gencode v27. Differential expression was assessed using DESeq2[69]. Comparison across and between control and IPF samples at days 4, 8, and 14 of ALI were performed with additional comparisons within control and IPF groups at different timepoints (i.e., control day 4 vs control day 8). Benjamini–Hochberg false discovery rate-adjusted $p$ values < 0.05 were considered significant. Pathway enrichment analyses were performed in Enrichr[70,71] and network analysis was done in Network Analyst[72] using the tissue-specific (lung) interactome dataset, only the largest network was considered.

**Poly(ethylene glycol)-norbornene (PEG-NB) synthesis.** Norbornene-functionalized poly(ethylene glycol) was synthesized following established protocols[73]. Briefly, PEG-hydroxyl (PEG-OH; MW 40 kg/mol, 8-arm, hexaglycerol core; JenKemUSA) was combined with a 5× molar ratio of pyridine (Fisher Scientific) and a 0.5× molar ratio of 4-(dimethylamino)pyridine (DMAP; Sigma-Aldrich) with respect to PEG hydroxyls in dichloromethane (DCM; Fisher Scientific). In a separate flask, a 10× molar ratio of 5-norbonene-2-carboxylic acid (NBCA; Sigma-Aldrich) and a 5× molar ratio of N,N′-dicyclohexylcarbodiimide (DCC; Sigma-Aldrich) with respect to PEG hydroxyls were added to DCM and stirred for 30 min at room temperature under argon (Airgas). The solution of PEG-OH and pyridine was added to the solution of NBCA and DCC and stirred for 2 days under argon. The reacted PEG-NB solution was then filtered through Celite 545 (EMD Millipore) soaked with DCM in a fritted glass funnel under vacuum and washed three times with 5% sodium bicarbonate (Sigma-Aldrich) in a separatory funnel. PEG-NB solution was precipitated in ice-cold diethyl ether (Fisher Scientific). The solution was then centrifuged at −10 °C and 4816 $g$ for 10 min. The supernatant was decanted off, the precipitate resuspended in ice-cold diethyl ether, and the centrifugation process repeated twice. The product was dried under vacuum overnight at room temperature. This product was then dialyzed in deionized water for 4 days at room temperature with daily water changes. Dialyzed product was then frozen and lyophilized until dry.

**Photoinitiator synthesis.** Lithium phenyl (2,4,6-trimethylbenzoyl) phosphinate (LAP) photoinitiator was synthesized utilizing established methods[73]. Briefly, equimolar amounts of dimethyl phenylphosphonite (Alfa Aesar) and 2,4,6-trimethylbenzoylphosphonite (Acros Organics) were added to a round-bottom flask and stirred overnight at room temperature under argon. A 4× molar ratio of lithium bromide (Acros Organics) with respect to dimethyl phenylphosphonite was dissolved in 2-butanone (Acros Organics) in a separate vessel. The reaction was stirred until all components dissolved and this solution was added to the round-bottom flask containing dimethyl phenylphosphonite and 2,3,6-trimethylbenzoylphosphonite. This reaction was heated to 50 °C until a precipitate formed (about 10 min). The reaction solution was cooled to room temperature over the course of an hour, filtered through filter paper moistened with 2-butanone in a Buchner funnel under vacuum, and rinsed thrice with 2-butanone. LAP product was dried under vacuum overnight at room temperature, dissolved in deionized water, frozen, and lyophilized until dry.

**Hydrogel fabrication.** PEG-NB hydrogels were polymerized using established methods[73]. The hydrogel prepolymer solution was created by combining 5 wt% 8-arm, 40 kg/mol PEG-NB, dithiothreitol (DTT; Sigma-Aldrich), 0.05 wt% LAP, and 2 mM of CGRGDS peptide in sterile phosphate buffered saline (PBS, pH = 7.2; Life Technologies). Glass cover slips (18 mm; Fisher Scientific) were silanated with (3-mercaptopropyl)trimethoxysilane (Acros Organics) using a liquid-deposition technique[74]. Hydrogels were photopolymerized in 90 µL drops placed between hydrophobic glass slides treated with SigmaCote (Sigma-Aldrich) and silanated 18 mm cover slips under 365 nm light at 10 mW/cm² for 5 min. Hydrogels were then swollen in complete medium at 37 °C overnight prior to use in experiments.

**Hydrogel mechanical characterization.** The elastic modulus (E) of swollen hydrogel samples was measured utilizing established methods[73]. Briefly, cylindrical hydrogel samples (volume = 50 uL) were polymerized between hydrophobic glass slides separated by a 0.5 mm silicone gasket, allowed to swell to equilibrium in PBS, and then cut to 8 mm diameter with a circular punch. The storage (G′) and loss moduli (G″) were measured for four replicates by loading hydrogel samples between an 8 mm parallel plate and a Peltier plate heated to 37 °C on a Discovery Hybrid Rheometer 2 (TA Instruments). The gap distance between the plate and the geometry was adjusted until the G′ measurement plateaued, and this percent compression was used for all samples[75]. Samples were subjected to shear at 1% strain through a dynamic angular frequency range of 0.1–100 rad/s. Rubber

elasticity theory was used to calculate E from G′ assuming a Poisson's ratio of 0.5, corresponding to an incompressible material.

**Statistics and reproducibility.** Data were either analyzed in PRISM or MATLAB with custom scripts. Statistical significance was determined via either an ANOVA or multiple $T$-tests. For the ANOVA a Kruskal–Wallis test was performed with post-hoc Dunn test ($p < 0.05$ was considered statistically significant). For multiple $t$-tests analysis was corrected using the Holm–Sidak test with statistical significance $p < 0.05$. For single $t$-tests a Mann–Whitney test was used with statistical significance $p < 0.05$. Airway epithelial experiments were performed with at least $n \geq 3$ donors, whereby cells were grown in the same methods. A $n \geq 3$ wells was used per donor for each experimental condition. Supplementary Table 1 describes donor information as well as the experiment where the cells were utilized. Fibroblast experiments utilized $n = 4$ donors for all experiments and each condition was replicated over at least three independent hydrogels. Cell dynamics experiments were averaged over four positions per well, with a minimum of three wells per treatment condition. Experiments were repeated at least three independent times and results were averaged over all repeats.

**Reporting summary.** Further information on research design is available in the Nature Research Reporting Summary linked to this article.

## Data availability

All data are available in the main text or the supplementary materials. All findings from the study are available upon request to the corresponding author. Source data are provided with this paper. RNA-Seq data has been deposited to the NCBI Gene Expression Omnibus and are accessed at GSE17600. Source data are provided with this paper.

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

## Acknowledgements

The authors thank Jonathan Cardwell for his assistance in aligning the RNA-Sequencing data and the Advanced Light Microscopy Core part of NeuroTechnology Center at

University of Colorado Anschutz Medical Campus. This research was supported by grants from the NHLBI (R01-HL097163, P01-HL092870, UH3-HL123442, R01-HL153096, 5T32-HL072738, RO1HL148152, P30ES000002), NINDS (P0-NS048154), NIDDK (P30-DK116073), NIA (T32-AG000279), NSF CAREER (1941401), PR192068, and DoD (W81XWH-17-1-0597).

## Author contributions

I.T.S., J.E.M., J.-A.P., C.M.M., E.D., and D.A.S. designed experiments, I.T.S., J.E.M., D.D-H., and E.D. performed experiments, I.T.S. and B.J.S. developed code, I.T.S., B.J.S., and D.A.S performed formal analysis, H.W.C, C.M.M., I.V.Y., B.J.S., and D.A.S. contributed resources, I.T.S., I.V.Y., and B.J.S. curated data, I.T.S. and D.A.S. drafted the original paper, and I.T.S., J.E.M., D.D-H., H.W.C., J.-A.P., C.M.M., I.V.Y., B.J.S., E.D., and D.A.S. edited the paper.

## Competing interests

DAS is the founder and chief scientific officer of Eleven P15, Inc., a company focused on the early diagnosis and treatment of pulmonary fibrosis. IVY is a consultant to Eleven P15, Inc. No other authors declare any competing interests.
