## [Peer Review File · Nature Communications]

REVIEWER COMMENTS

Reviewer #1 (Remarks to the Author):

If true, this report represents a major advance in our understanding of idiopathic pulmonary fibrosis. In a series of innovative experiments, it brings together cell biology, biophysics and genetics to show the unexpected finding that the pathogenesis of IPF may hinge upon epithelial unjamming in the small airways of the lung. Both with regard to the locus of this event (small airway rather than alveolus and matrix) and with regard to mechanism (epithelial unjamming), this comes as a surprise. For the most part, the writing of the main text is strong, elegant, and clear. The writing of the figure legends, by contrast, is underdeveloped, cryptic, and, by way of crucial omissions, places before the reader obstacles and confusion, about which I say more below. Each legend needs to be rewritten from scratch in its entirety and with an eye toward telling a self-contained story.

Major:

Major strengths of this report are the manner in which it seems to delineate in a rather definitive fashion 1) small airway from large airway, 2) IPF from control, and 3) jammed versus unjammed. It is stunning that in each of these dimensions the story is pretty much black or white. This paper establishes that in the distal airway: 1) ERBB-YAP activation is sufficient to induce unjamming; 2) ERBB-YAP axis specifically regulates IPF unjamming; 3) unjamming is responsive to genetic risk variants and is disease specific. Each is a major finding.

The reason for my qualifier above “if true”—and potentially a fatal flaw— is that Fig. 1 is ambiguous. Moreover, depending on information that is missing from both the figure legend and the main text, Fig. 1 may contain a major inconsistency. For Fig 1 panels D, E, F, G, it is not made clear if these data are from small airways, large airways or both. On my first read I inferred that the authors meant larger airways. But if that is the case, then Fig. 1 panels D-G are totally inconsistent with Fig. 1 panels I, J, K, L. On the other hand, if Fig. 1 panels D,E,F,G refer to smaller airways, then this crucial fact is left unstated. Moreover, I could find nowhere in the manuscript how cells were extracted differentially from the smaller airways as opposed to the larger ones. Given that this differential extraction impacts one of the major findings, the absence of this information is striking.

Similarly, the analysis of RNAseq fails to specify the airway size to which the analysis pertains.

As regards unjamming and the analyses presented here of KEGG, gene ontology, networks, and protein-protein interactions, striking by their omission are any references at all to, or contrasts

between, previously reported RNAseq analyses of unjamming using closely similar analytical methods in these very same epithelial cell types but with different unjamming stimuli (mechanical compression or asthma) (Kilic et al, Sci. Rep, 2020; de Marzio et al, bioRxiv, 2020). In the interests of validation of the current analysis as well as learning about mechanistic differences between various unjamming stimuli, not to mention good scholarship, contrasts to these earlier works would seem to be essential.

Minor:

46-47 "To understand how disease status alters this tissue layer, we investigated whether the unjammed to-jammed transition (UJT) was altered in the epithelia".

To avoid confusion, you may want alert the reader at this juncture that in the literature some authors use "UJT" to mean the transition from a jammed, solid-like, non-motile phase to an unjammed, fluid-like, motile phase, which is precisely the opposite of the usage here.

Unless I am mistaken, Jin-Ah Park provided instrumental help in the early stages of this study. If not acknowledged by co-authorship, I was surprised and a bit disappointed to see that her name did not at least appear in acknowledgments.

Jeffrey Fredberg

Reviewer #2 (Remarks to the Author):

I was asked to provide feedback exclusively on those technical aspects related to hydrogel synthesis and characterization. The hydrogels used in this submission were crosslinked by 8-arm poly(ethylene glycol)-norbornene (PEG-NB), DTT, and immobilized with 2 mM of CRGDS peptide for permitting cell adhesion. The cells were seeded on top of the hydrogels, which were fabricated to around $E=3\text{kPa}$.

I find no novelty in terms of macromer/hydrogel synthesis and characterization as this is a very well-established gel chemistry. The authors did not study how hydrogel mechanics affect cell behavior, although it was mentioned that $E=3\text{kPa}$ was selected to "prevent artificial activation of HLFs". Nonetheless, it will be nice to see how hydrogel mechanics affect HLF activation under the influence of YAP/EGFR inhibitors.

Reviewer #3 (Remarks to the Author):

Overview: This is a well written and data-rich paper of interest to the field. Major novel findings include the maintenance of abnormal epithelial cell behavior of IPF epithelial cells in vitro and the distinct interaction of IPF cells with fibroblasts that may contribute to fibrogenesis in IPF. Data demonstrating YAP activation in IPF epithelial cells in vivo, and in vitro data regarding the primary importance of YAP signaling in cell migration, "piling", and gene expression confirming previous studies. Experiments demonstrating that signals from IPF epithelial cells influence fibroblasts that may contribute to fibrotic remodeling are of considerable interest to the field. The authors have carefully demonstrated the YAP-ERRB signaling network with inhibitors supporting the important role of YAP.

Major concerns:

1. A major conclusion is the abstract states "this biophysical state and signaling axis directs activation in the underlying mesenchyme". However, direct links between "biophysics", stress/stretch, on the secretion of ligands by the epithelium that in turn may regulate the mesenchyme are not supported by direct experimentation. Likewise, conclusions that cell shape affects the underlying mesenchyme to activate fibrosis merits direct experimentation or modification of the statement.
2. The authors use "jamming/un-jamming" as a mode of migration of a sheet of cells. For example, effects of YAP on cell migration, shape change, differentiation, "piling" were previously reported in studies of YAP in IPF, in mouse models and in vitro. Careful referencing and integration of previous work should be included in the present work. The authors provide evidence that changes in cell behavior and shapes distinct from EMT. Alterations in PCP components and cell shape were previously described in IPF tissue. Present findings confirm previous work demonstrating the effects of YAP and its inhibition on cell shape and migration of airway epithelial cells.

Specific Comments:

1. Figure legends, particularly Figure 1, "error bars represent 95% confidence interval" is written several times. Since error bars throughout each figure appear to be represented as such, the authors might write the statement at the end of each figure legend.
2. Figure 2, Panel A: DNA staining is very weak in IPF images and as such, it is unclear whether AREG is nuclear.

3. Figure 2, Panel B: Staining in control nuclei appear out of focus compared to the other images, making it difficult to identify nuclear localization.
4. Increased expression of Muc5B and Krt5 is well established in IPF epithelial cells and as such should be referenced; e.g. multiple single-cell sequencing papers (Xu, Kropski, Kaminski, Dressen) have shown this.
5. As CTGF is also regulated by TGFB; perhaps an analysis of AXL may be more YAP specific, in previous groups identified increased AXL in IPF epithelial cells and it is expressed in HBEC cells.
6. Figure S6, A-C: Arrows would assist in the identification of “co-localization”.
7. Figure S6 Panel H: It is not clear what the white arrows are pointing to, is these FOXJ1 positive nuclei, or is there nuclear YAP? The positive YAP staining seems to be apically cytoplasmic and its nuclear cell type.
8. Figure S9: All treatments appear to block YAP nuclear localization; thus a control image might be added.
9. Figure S9/S10: Was cell viability assessed? Confluence appears to vary among treatments.
10. Muc5B variants: Interestingly, cells with mutations in MUC5B treated with XMU or AREG are associated with increased cell speed. The authors might discuss how these processes might be associated or correlated. How does MUC5B, a secreted cell product, influence YAP induced cell migration?
11. It is difficult to assess the number of replicates or what each puncta represents in most graphs. Methods state that each experiment used 3 or more donors with 3 or more wells each, but several of the graphs have more than 9 puncta (Figure 3 panels F/G) or only one punctum with a confidence interval. Providing clear Ns and replicates in figure legends would be helpful.

REVIEWER COMMENTS

Reviewer #1 (Remarks to the Author):

5 If true, this report represents a major advance in our understanding of idiopathic pulmonary fibrosis. In
a series of innovative experiments, it brings together cell biology, biophysics and genetics to show the
unexpected finding that the pathogenesis of IPF may hinge upon epithelial unjamming in the small
airways of the lung. Both with regard to the locus of this event (small airway rather than alveolus and
matrix) and with regard to mechanism (epithelial unjamming), this comes as a surprise. For the most
10 part, the writing of the main text is strong, elegant, and clear. The writing of the figure legends, by
contrast, is underdeveloped, cryptic, and, by way of crucial omissions, places before the reader
obstacles and confusion, about which I say more below. Each legend needs to be rewritten from scratch
in its entirety and with an eye toward telling a self-contained story.

15 Major:
Major strengths of this report are the manner in which it seems to delineate in a rather definitive
fashion 1) small airway from large airway, 2) IPF from control, and 3) jammed verses unjammed. It is
stunning that in each of these dimensions the story is pretty much black or white. This paper establishes
that in the distal airway: 1) ERBB-YAP activation is sufficient to induce unjamming; 2) ERBB-YAP axis
20 specifically regulates IPF unjamming; 3) unjamming is responsive to genetic risk variants and is disease
specific. Each is a major finding.

The reason for my qualifier above “if true”—and potentially a fatal flaw— is that Fig. 1 is ambiguous.
Moreover, depending on information that is missing from both the figure legend and the main text, Fig.
25 1 may contain a major inconsistency. For Fig 1 panels D, E, F, G, it is not made clear if these data are
from small airways, large airways or both. On my first read I inferred that the authors meant larger
airways. But if that is the case, then Fig. 1 panels D-G are totally inconsistent with Fig. 1 panels I, J, K, L.
On the other hand, if Fig. 1 panels D,E,F,G refer to smaller airways, then this crucial fact is left unstated.
Moreover, I could find nowhere in the manuscript how cells were extracted differentially from the
30 smaller airways as opposed to the larger ones. Given that this differential extraction impacts one of the
major findings, the absence of this information is striking.

Similarly, the analysis of RNAseq fails to specify the airway size to which the analysis pertains.

35 As regards unjamming and the analyses presented here of KEGG, gene ontology, networks, and protein-
protein interactions, striking by their omission are any references at all to, or contrasts between,
previously reported RNAseq analyses of unjamming using closely similar analytical methods in these
very same epithelial cell types but with different unjamming stimuli (mechanical compression or
asthma) (Kilic et al, Sci. Rep, 2020; de Marzio et al, bioRxiv, 2020). In the interests of validation of the
40 current analysis as well as learning about mechanistic differences between various unjamming stimuli,
not to mention good scholarship, contrasts to these earlier works would seem to be essential.

Minor:

45 46-47 “To understand how disease status alters this tissue layer, we investigated whether the
unjammed to-jammed transition (UJT) was altered in the epithelia”.
To avoid confusion, you may want alert the reader at this juncture that in the literature some authors
use “UJT” to mean the transition from a jammed, solid-like, non-motile phase to an unjammed, fluid-
like, motile phase, which is precisely the opposite of the usage here.

50

Unless I am mistaken, Jin-Ah Park provided instrumental help in the early stages of this study. If not acknowledged by co-authorship, I was surprised and a bit disappointed to see that her name did not at least appear in acknowledgments.

Jeffrey Fredberg

55

Reviewer #2 (Remarks to the Author):

60 I was asked to provide feedback exclusively on those technical aspects related to hydrogel synthesis and characterization. The hydrogels used in this submission were crosslinked by 8-arm poly(ethylene glycol)-norbornene (PEG-NB), DTT, and immobilized with 2 mM of CRGDS peptide for permitting cell adhesion. The cells were seeded on top of the hydrogels, which were fabricated to around $E=3\text{kPa}$.

65 I find no novelty in terms of macromer/hydrogel synthesis and characterization as this is a very well-established gel chemistry. The authors did not study how hydrogel mechanics affect cell behavior, although it was mentioned that $E=3\text{kPa}$ was selected to "prevent artificial activation of HLFs". Nonetheless, it will be nice to see how hydrogel mechanics affect HLF activation under the influence of YAP/EGFR inhibitors.

70

75

Reviewer #3 (Remarks to the Author):

80 Overview: This is a well written and data-rich paper of interest to the field. Major novel findings include the maintenance of abnormal epithelial cell behavior of IPF epithelial cells in vitro and the distinct interaction of IPF cells with fibroblasts that may contribute to fibrogenesis in IPF. Data demonstrating YAP activation in IPF epithelial cells in vivo, and in vitro data regarding the primary importance of YAP signaling in cell migration, "piling", and gene expression confirming previous studies. Experiments demonstrating that signals from IPF epithelial cells influence fibroblasts that may contribute to fibrotic remodeling are of considerable interest to the field. The authors have carefully demonstrated the YAP-ERRB signaling network with inhibitors supporting the important role of YAP.

85 Major concerns:

90 1. A major conclusion is the abstract states "this biophysical state and signaling axis directs activation in the underlying mesenchyme". However, direct links between "biophysics", stress/stretch, on the secretion of ligands by the epithelium that in turn may regulate the mesenchyme are not supported by direct experimentation. Likewise, conclusions that cell shape affects the underlying mesenchyme to activate fibrosis merits direct experimentation or modification of the statement.

95 2. The authors use "jamming/un-jamming" as a mode of migration of a sheet of cells. For example, effects of YAP on cell migration, shape change, differentiation, "piling" were previously reported in studies of YAP in IPF, in mouse models and in vitro. Careful referencing and integration of previous work should be included in the present work. The authors provide evidence that changes in cell behavior and shapes distinct from EMT. Alterations in PCP components and cell shape were previously described in IPF tissue. Present findings confirm previous work demonstrating the effects of YAP and its inhibition on cell shape and migration of airway epithelial cells.

100 Specific Comments:

105 1. Figure legends, particularly Figure 1, "error bars represent 95% confidence interval" is written several times. Since error bars throughout each figure appear to be represented as such, the authors might write the statement at the end of each figure legend.

110 2. Figure 2, Panel A: DNA staining is very weak in IPF images and as such, it is unclear whether AREG is nuclear.

115 3. Figure 2, Panel B: Staining in control nuclei appear out of focus compared to the other images, making it difficult to identify nuclear localization.

4. Increased expression of Muc5B and Krt5 is well established in IPF epithelial cells and as such should be referenced; e.g. multiple single-cell sequencing papers (Xu, Kropski, Kaminski, Dressen) have shown this.

5. As CTGF is also regulated by TGF β ; perhaps an analysis of AXL may be more YAP specific, in previous groups identified increased AXL in IPF epithelial cells and it is expressed in HBEC cells.

6. Figure S6, A-C: Arrows would assist in the identification of "co-localization".

7. Figure S6 Panel H: It is not clear what the white arrows are pointing to, is these FOXJ1 positive nuclei, or is there nuclear YAP? The positive YAP staining seems to be apically cytoplasmic and its nuclear cell type.

8. Figure S9: All treatments appear to block YAP nuclear localization; thus a control image might be added.

9. Figure S9/S10: Was cell viability assessed? Confluence appears to vary among treatments.

120 10. Muc5B variants: Interestingly, cells with mutations in MUC5B treated with XMU or AREG are associated with increased cell speed. The authors might discuss how these processes might be associated or correlated. How does MUC5B, a secreted cell product, influence YAP induced cell migration?

11. It is difficult to assess the number of replicates or what each puncta represents in most graphs. Methods state that each experiment used 3 or more donors with 3 or more wells each, but several of the graphs have more than 9 puncta (Figure 3 panels F/G) or only one punctum with a confidence interval. Providing clear Ns and replicates in figure legends would be helpful.

Response to Editor:

130

We thank the reviewers for their thoughtful and constructive comments. We have worked to improve the clarity of our manuscript and address reviewer concerns. Our point-by-point responses to the reviewer comments to the manuscript are listed below.

135

Reviewer #1 (Remarks to the Author):

140

If true, this report represents a major advance in our understanding of idiopathic pulmonary fibrosis. In a series of innovative experiments, it brings together cell biology, biophysics and genetics to show the unexpected finding that the pathogenesis of IPF may hinge upon epithelial unjamming in the small airways of the lung. Both with regard to the locus of this event (small airway rather than alveolus and matrix) and with regard to mechanism (epithelial unjamming), this comes as a surprise. For the most part, the writing of the main text is strong, elegant, and clear. The writing of the figure legends, by contrast, is underdeveloped, cryptic, and, by way of crucial omissions, places before the reader obstacles and confusion, about which I say more below. Each legend needs to be rewritten from scratch in its entirety and with an eye toward telling a self-contained story.

145

Major:

150

Major strengths of this report are the manner in which it seems to delineate in a rather definitive fashion 1) small airway from large airway, 2) IPF from control, and 3) jammed versus unjammed. It is stunning that in each of these dimensions the story is pretty much black or white. This paper establishes that in the distal airway: 1) ERBB-YAP activation is sufficient to induce unjamming; 2) ERBB-YAP axis specifically regulates IPF unjamming; 3) unjamming is responsive to genetic risk variants and is disease specific. Each is a major finding.

155

Comment: The reason for my qualifier above “if true”—and potentially a fatal flaw— is that Fig. 1 is ambiguous. Moreover, depending on information that is missing from both the figure legend and the main text, Fig. 1 may contain a major inconsistency. For Fig 1 panels D, E, F, G, it is not made clear if these data are from small airways, large airways or both. On my first read I inferred that the authors meant larger airways. But if that is the case, then Fig. 1 panels D-G are totally inconsistent with Fig. 1 panels I, J, K, L. On the other hand, if Fig. 1 panels D,E,F,G refer to smaller airways, then this crucial fact is left unstated. Moreover, I could find nowhere in the manuscript how cells were extracted differentially from the smaller airways as opposed to the larger ones. Given that this differential extraction impacts one of the major findings, the absence of this information is striking.

160

165

Response: This is a critical point and oversight on our behalf. We thank the reviewer for pointing this out. Fig 1 panels D-G (control vs. IPF experiments) are referring specifically to cells isolation from distal airways consistent with the findings from Fig 1 I-L (IPF proximal vs. distal airway experiments). Here we have defined distal airways as airways with a diameter ≤ 2 mm or for bronchial brushings the 6th or 7th airway generation. We have modified the body text, methods, and figures to make this point abundantly clear.

170

Text: We have significantly modified the text referring to figure 1 to enhance clarity. As now described, we say that cells utilized for Fig 1. D-G are from bronchial brushes of distal airways. We have also added in the body of the text that the proximal vs. distal experiment, Fig 1. I-L are from dissected airways. Additionally, we have emphasized that the only proximal (or tracheobronchial cells) utilized were done so in Fig 1. I-L.

Methods: We have added information on the airway diameter for epithelial cell isolation as well as the similarity of culture conditions regardless of isolation method prior to experimentation.

175

Figure: We have modified figure 1 to refer to cells and regions as proximal and distal. Also, to enhance clarity, we have added two additional columns to supplemental table 1 that detail if the cells were isolated from a proximal or distal airway as well as the method of isolation. We have also rewritten all main figure legends.

180

Comment: Similarly, the analysis of RNAseq fails to specify the airway size to which the analysis pertains.

Response: This again is a critical and important point. We have again modified the text to clarify that the epithelial cells sequenced were from distal airway brushings. We have also modified supplemental table 1 with the addition of a column detailing the region where the epithelial samples were isolated.

185

Text: As now described in the text we have added that the RNA-seq was performed on distal airway epithelial cells from control of IPF donors. We have also amended supplemental table 1 to include airway location as it relates to experimentation.

190

Comment: As regards unjamming and the analyses presented here of KEGG, gene ontology, networks, and protein-protein interactions, striking by their omission are any references at all to, or contrasts between, previously reported RNAseq analyses of unjamming using closely similar analytical methods in these very same epithelial cell types but with different unjamming stimuli (mechanical compression or asthma) (Kilic et al, Sci. Rep, 2020; de Marzio et al, bioRxiv, 2020). In the interests of validation of the current analysis as well as learning about mechanistic differences between various unjamming stimuli, not to mention good scholarship, contrasts to these earlier works would seem to be essential.

195

Response: Thank you for raising this point. We agree the comparison across datasets is critically important to deepening the understanding of gene/protein regulators of the unjammed state. Kilic and Marzio both present data from a mechanical induced unjammed state whereas we present data from a differentiation unjammed/jammed state. We believe that there are significant and important differences between the two unjammed states as such we have made the following text changes:

200

Text: We have modified the body of the text to include comparisons to Kilic et al and Marzio et al.

205

Minor:

Comment: 46-47 “To understand how disease status alters this tissue layer, we investigated whether the unjammed to-jammed transition (UJT) was altered in the epithelia”.

210

To avoid confusion, you may want alert the reader at this juncture that in the literature some authors use “UJT” to mean the transition from a jammed, solid-like, non-motile phase to an unjammed, fluid-like, motile phase, which is precisely the opposite of the usage here.

Response: We thank the reviewer for this point – to avoid any confusion we have shifted our use of the UJT to mirror the previously published form in the field for consistency.

215

Comment: Unless I am mistaken, Jin-Ah Park provided instrumental help in the early stages of this study. If not acknowledged by co-authorship, I was surprised and a bit disappointed to see that her name did not at least appear in acknowledgments.

220

Response: We thank the reviewer for raising this point. Dr. Park was involved in discussing the initiation experimental approaches to our research, and we neglected to properly acknowledge her for her significant contributions. After further discussion with Dr. Park, she is now included as a co-author on the manuscript. Dr. Park has also provided critical revisions to the current manuscript text.

225

Jeffrey Fredberg

Reviewer #2 (Remarks to the Author):

230 I was asked to provide feedback exclusively on those technical aspects related to hydrogel synthesis and characterization. The hydrogels used in this submission were crosslinked by 8-arm poly(ethylene glycol)-norbornene (PEG-NB), DTT, and immobilized with 2 mM of CRGDS peptide for permitting cell adhesion. The cells were seeded on top of the hydrogels, which were fabricated to around $E=3\text{kPa}$.

235 Comment: I find no novelty in terms of macromer/hydrogel synthesis and characterization as this is a very well-established gel chemistry. The authors did not study how hydrogel mechanics affect cell behavior, although it was mentioned that $E=3\text{kPa}$ was selected to "prevent artificial activation of HLFs". Nonetheless, it will be nice to see how hydrogel mechanics affect HLF activation under the influence of YAP/EGFR inhibitors.

240 **Response:** We thank the reviewer for their comments, and we agree that this is an established method in the field. Our specific goal for utilization of a hydrogel system was to prevent artificial activation of primary human lung fibroblasts from tissue culture plastic. Our results in Fig S14 demonstrated a baseline activation of the cells preventing a full assessment of the epithelial induction of the fibroblast phenotype. Through the utilization of a common hydrogel system, we were able to identify robust differences in fibroblast activation from epithelial media treatments.

245
250 We agree that mechanical activation of fibroblasts is a critically important scientific question as it allows for modeling of an *in vivo* stiffness *in vitro* and that the mechanisms regulating this activation deepens our understanding of fibrosis and cell biology. However, examining this relationship was not within the scope of the current manuscript. Several groups have pioneered this field, specifically in the context of pulmonary fibrosis as it pertains to EGFR or YAP activation (Liu et al, AJP 2015, Haak et al, Sci Transl Med 2019, Zhou et al, J Biol Chem 2012) and have been referenced appropriately.

Reviewer #3 (Remarks to the Author):

255

Overview: This is a well written and data-rich paper of interest to the field. Major novel findings include the maintenance of abnormal epithelial cell behavior of IPF epithelial cells in vitro and the distinct interaction of IPF cells with fibroblasts that may contribute to fibrogenesis in IPF. Data demonstrating YAP activation in IPF epithelial cells in vivo, and in vitro data regarding the primary importance of YAP signaling in cell migration, "piling", and gene expression confirming previous studies. Experiments demonstrating that signals from IPF epithelial cells influence fibroblasts that may contribute to fibrotic remodeling are of considerable interest to the field. The authors have carefully demonstrated the YAP-ERRB signaling network with inhibitors supporting the important role of YAP.

260

265

Major concerns:

Comment: A major conclusion is the abstract states "this biophysical state and signaling axis directs activation in the underlying mesenchyme". However, direct links between "biophysics", stress/stretch, on the secretion of ligands by the epithelium that in turn may regulate the mesenchyme are not supported by direct experimentation. Likewise, conclusions that cell shape affects the underlying mesenchyme to activate fibrosis merits direct experimentation or modification of the statement.

270

Response: We thank the reviewer for their comments. We understand the concern with the assertion that the biophysical state alone (as induced by AREG or XMU treatment) directly results in a pro-fibrotic phenotype. We agree that further experimentation would be needed to establish causation, specifically. As currently presented, we believe there is a strong correlation between the biophysical state and fibroblast activation. We believe, to fully establish the biophysical-directed fibroblast activation would need an alternative model of epithelial unjamming, specifically arising purely from a mechanical source. As currently presented and with current methods, we are unable to decouple the signaling and biophysical aspects of the model as it relates to fibroblast activation. While we understand the specific signaling regulators, we currently lack the methods for establishing a purely mechanical approach of unjamming which would disentangle the signaling and biophysical contributions to fibroblast activation.

275

280

Such models have been established by other labs (i.e., Park et al., 2015 and Mitchel et al., 2020) in the context of mechanical models of asthma pathogenesis. This model applies a physiologic compressive force to the apical surface of the epithelia to induce an unjammed state. However, the relevance of this model to pulmonary fibrosis would need to be studied and established outside of this study. Interestingly, however, Swartz et al., 2001 PNAS demonstrated that mechanical pressure to the epithelia induces a pro-fibrotic outcome in human lung fibroblast. But, the link to jamming still remains to be fully established.

285

290

Additionally, recent work from these groups have identified that compression-induced unjamming elicits a distinct gene/protein signature that is dissimilar to the signature identified in the present study (Kilic et al., 2020 and de Marzio et al., 2020). However, it is an important point that future studies should include model development to understand how mechanical forces, alone, can induce unjamming in epithelial cells to elicit a phenotype similar to that found in pulmonary fibrosis epithelia. We have made the following text changes to address the reviewer's concern.

295

Text: We have modified the body of the text (line: 30).

300 Comment: The authors use "jamming/un-jamming" as a mode of migration of a sheet of cells. For example, effects
of YAP on cell migration, shape change, differentiation, "piling" were previously reported in studies of YAP in IPF, in
mouse models and in vitro. Careful referencing and integration of previous work should be included in the present
work. The authors provide evidence that changes in cell behavior and shapes distinct from EMT. Alterations in PCP
305 components and cell shape were previously described in IPF tissue. Present findings confirm previous work
demonstrating the effects of YAP and its inhibition on cell shape and migration of airway epithelial cells.

**Response: We thank the reviewer for their comments and agree that aspects of YAP tissue biology have been
demonstrated by previous groups. As such, we have added appropriate citations to works from Xu et al, JCI
Insight 2016, Gokey et al, JCI Insight 2018, Lange et al, and J. Mol. Cell Biol 2015. These previous works have
310 provided an experimental and conceptual framework of how to understand airway epithelial YAP signaling at
homeostasis and in disease.**

**Our novel contributions to YAP signaling in the airway epithelium arise through the link of YAPs necessity to the
unjammed state as well establishing an ERBB/AREG dependent mechanism. Additionally, we believe the novel
315 findings of YAP will be particularly interesting to those investigating epithelial unjamming as YAP is classically
thought of as a mechanotransducing molecule.**

Specific Comments:

320 Comment: Figure legends, particularly Figure 1, "error bars represent 95% confidence interval" is written several
times. Since error bars throughout each figure appear to be represented as such, the authors might write the
statement at the end of each figure legend.

**Response: We agree that figure legends were unnecessarily repetitive. As such, we have rewritten all figure
325 legends to enhance clarity. We have included a concluding statement at the end of each figure legend stating
that error bars throughout the figure correspond to 95% confidence intervals instead of repeating throughout
legend.**

330 Comment: Figure 2, Panel A: DNA staining is very weak in IPF images and as such, it is unclear whether AREG is
nuclear.

**Response: Thank you for this comment – as such we have included an updated figure with increased
DAPI/nuclear contrast.**

335 Comment: Figure 2, Panel B: Staining in control nuclei appear out of focus compared to the other images, making it
difficult to identify nuclear localization.

**Response: We thank the reviewer for this comment and agree that the control HBEC YAP staining is difficult to
visualize. As such, we have altered figure 2 panel b to include only YAP staining. In doing so, we believe that the
340 control HBEC cytoplasmic staining can be easily contrasted to the IPF HBEC nuclear staining.**

Comment: Increased expression of Muc5B and Krt5 is well established in IPF epithelial cells and as such should be
referenced; e.g. multiple single-cell sequencing papers (Xu, Kropski, Kaminski, Dressen) have shown this.

345 **Response: We appreciate the reviewer's comments. We agree that MUC5B and KRT5 increased expression in the IPF epithelium has been previous demonstrate by our group and others specifically at the gene and protein level. As such we have included appropriate citations to previous studies that have established these differences *in vivo*.**

350 Comment: As CTGF is also regulated by TGFB; perhaps an analysis of AXL may be more YAP specific, in previous groups identified increased AXL in IPF epithelial cells and it is expressed in HBEC cells.

355 **Response: We thank the reviewer for their comment, and we agree. We had not previously considered a how AXL might interact with our proposed mechanism, and we agree that there is merit to this axis. While our manuscript specifically describes an ERBB-YAP axis, it is possible that AXL heterodimerize with EGFR/ERBB2/ERBB3 or signals independently. While we did not see a gene level signature that would point us specifically to a role for AXL, future work should investigate the possible role of AXL/ERBB directed YAP activation in pulmonary fibrosis epithelial dysfunction especially as it relates to the unjammed state.**

360 Comment: Figure S6, A-C: Arrows would assist in the identification of "co-localization".

Response: We agree and have added arrows to the immunofluorescence images to guide the reader in identifying the cells where FOXJ1 and YAP are co-localized in the nucleus.

365 Comment: Figure S6 Panel H: It is not clear what the white arrows are pointing to, is these FOXJ1 positive nuclei, or is there nuclear YAP? The positive YAP staining seems to be apically cytoplasmic and its nuclear cell type.

Response: We agree the staining contrast is difficult to visualize. We have increased the image contrast and included arrows pointing specifically to YAP/FOXJ1 co-positive nuclei.

370 Comment: Figure S9: All treatments appear to block YAP nuclear localization; thus a control image might be added.

375 **Response: We agree – we have added both an untreated control and IPF image as a point of contrast for the other treatment conditions.**

Comment: Figure S9/S10: Was cell viability assessed? Confluence appears to vary among treatments.

380 **Response: Transepithelial electrical resistance was utilized as a proxy for cell viability for these experiments. Those data are present in Fig. S8. We noticed that there was no decrease in transepithelial electrical resistance after inhibitor treatment, but a gradual increase consisted with untreated IPF samples. Additionally, Fig. S8 shows that after inhibitor treatment, IPF HBECs regardless of inhibitor treatment status all increase barrier function for up to 144 hours after initiation of treatment. For this reason, we believe that there is no decrease in cell viability of loss of barrier function.**

385 Comment: Muc5B variants: Interestingly, cells with mutations in MUC5B treated with XMU or AREG are associated with increased cell speed. The authors might discuss how these processes might be associated or correlated. How does MUC5B, a secreted cell product, influence YAP induced cell migration?

390

Response: Thank you for this comment – and we agree that *MUC5B* variant status is an interesting phenotype as it relates to unjamming. Our current conceptual framework is that there is a shift in metabolism towards glycolysis. Previous groups have demonstrated metabolic shifts towards glycolysis during the unjammed state, cytoskeletal reorganization, and endoplasmic reticulum stress. We have added the following text changes to speculate how the promoter variant might be interacting with ERBB/YAP to potentiate the unjammed state.

395

Text: We have modified the text to include this important discussion.

Comment: It is difficult to assess the number of replicates or what each puncta represents in most graphs.

400

Methods state that each experiment used 3 or more donors with 3 or more wells each, but several of the graphs have more than 9 puncta (Figure 3 panels F/G) or only one punctum with a confidence interval. Providing clear Ns and replicates in figure legends would be helpful.

405

Response: Thank you for this comment and we agree. All main figure legends have been rewritten to included specific number of replicates utilized in each experiment.

REVIEWERS' COMMENTS

Reviewer #1 (Remarks to the Author):

These changes address my criticisms. This manuscript now makes for an important contribution to the literature.

Minor: The authors might want to do some proof-reading.

Reviewer #3 (Remarks to the Author):

The authors provide in vivo and in vitro data supporting an important role for a signaling axis by which YAP/ERBB influences the “jammed” transition in airway epithelia in IPF. Cell shape and migration are altered in IPF epithelial cells remaining in an “unjammed” state in a process mediated by EGFR and YAP as indicated by inhibitor studies. The jammed cells differentially influence fibroblast activation in a process inhibited by YAP and ERBB inhibitors. This study provides new insight into the abnormal activities of epithelial cells that may in turn influence fibrotic responses in IPF. The authors have addressed major concerns raised by the reviewers, have improved the Figures, methods, and referencing of prior work in the rebuttal.

REVIEWER COMMENTS

Reviewer #1 (Remarks to the Author):

5 These changes address my criticisms. This manuscript now makes for an important contribution to the literature.

Minor: The authors might want to do some proof-reading.

10

Reviewer #3 (Remarks to the Author):

15 The authors provide in vivo and in vitro data supporting an important role for a signaling axis by which YAP/ERBB influences the “jammed” transition in airway epithelia in IPF. Cell shape and migration are altered in IPF epithelial cells remaining in an “unjammed” state in a process mediated by EGFR and YAP as indicated by inhibitor studies. The jammed cells differentially influence fibroblast activation in a process inhibited by YAP and ERBB inhibitors. This study provides new insight into the abnormal activities of epithelial cells that may in turn influence fibrotic responses in IPF. The authors have addressed major concerns raised by the reviewers, have improved the Figures, methods, and
20 referencing of prior work in the rebuttal.

25

30

35

40

Response to Editor:

These changes address my criticisms. This manuscript now makes for an important contribution to the literature.

45

Minor: The authors might want to do some proof-reading.

Response: We thank reviewer 1 for their thoughtful and critical feedback, we have thoroughly reviewed the manuscript to ensure there are no errors present

Reviewer #3 (Remarks to the Author):

50

The authors provide in vivo and in vitro data supporting an important role for a signaling axis by which YAP/ERBB influences the “jammed” transition in airway epithelia in IPF. Cell shape and migration are altered in IPF epithelial cells remaining in an “unjammed” state in a process mediated by EGFR and YAP as indicated by inhibitor studies. The jammed cells differentially influence fibroblast activation in a process inhibited by YAP and ERBB inhibitors. This study provides new insight into the abnormal activities of epithelial cells that may in turn influence fibrotic responses in IPF. The authors have addressed major concerns raised by the reviewers, have improved the Figures, methods, and referencing of prior work in the rebuttal.

55

60

Response: Thank you